# Characteristics of CO_2_ and Energy-Saving Concrete with Porous Feldspar

**DOI:** 10.3390/ma13184204

**Published:** 2020-09-21

**Authors:** Jung-Geun Han, Jin-Woo Cho, Sung-Wook Kim, Yun-Suk Park, Jong-Young Lee

**Affiliations:** 1School of Civil and Environmental Engineering, Urban Design and Study, Chung-Ang University, Seoul 06974, Korea; jghan@cau.ac.kr (J.-G.H.); dbstjr9653@naver.com (Y.-S.P.); 2Construction Automation Research Center, Korea Institute of Civil Engineering and Building Technology (KICT), Goyang 10223, Korea; 3Geo Information Research Group Co., LTD., Busan 47598, Korea; suwokim@chol.com

**Keywords:** porous feldspar, activation, compressive strength, substitute material, energy saving concrete

## Abstract

In this study, to reduce the use of cement and sand, porous feldspar with excellent economic efficiency was used as a substitute in the heat storage concrete layer. When porous feldspar and four other silicate minerals were used as substitute materials for sand in cement mortar, the specimen with the porous feldspar exhibited approximately 16–63% higher compressive strength, thereby exhibiting a higher reactivity with cement compared to the other minerals. To compensate for the reduction in strength owing to the decreased cement content, mechanical and chemical activation methods were employed. When the specific surface area of porous feldspar was increased, the unit weight was reduced by approximately 30% and the compressive strength was increased by up to 90%. In addition, the results of the thermal diffusion test confirmed that thermal diffusion increased owing to a reduction in the unit weight; the heat storage characteristics improved owing to the better porosity of feldspar. When chemical activation was performed after reducing the cement content by 5% and replacing the sand with porous feldspar, the compressive strength was found to be approximately twice that of an ordinary cement mortar. In a large-scale model experiment, the heat storage layer containing the porous feldspar exhibited better heat conduction and heat storage characteristics than the heat storage layer composed of ordinary cement mortar. Additionally, energy savings of 57% were observed.

## 1. Introduction

One of the characteristics of South Korea’s housing culture is that hot water is circulated through the piping underneath the floor. Fossil fuels, such as LNG (liquefied natural gas), LPG (liquefied petroleum gas), and coal, are mainly used as the heat sources for the hot water supply, and cement is mostly used as a flooring material. Cement, which is based on carbonate minerals, has continuously generated debates about human health risks, such as sick house syndrome and atopy, owing to heavy metal release and high pH [1]. In South Korea, the energy target management system and the emissions trading scheme were introduced based on the 21st United Nations Framework Convention on Climate Change, and efforts are being made to reduce greenhouse gas emissions [2]. Therefore, a method for reducing the use of cement, which emits 700 kg of carbon dioxide per ton, is required [3]. In construction, fly ash, which is the residue of the thermal power generation process, is used as a substitute for cement. The mixture of fly ash and cement has been used as roadbed and fill material [4,5,6,7,8]. When substitute materials are mixed with cement, compressive strength characteristics vary depending on the mixing ratio. To prevent the reduction in strength owing to the decreased cement content, studies have been conducted on activation methods for increasing the reactivity of substitute materials. Furthermore, studies on eco-friendly materials and the reduction in the use of cement have been conducted of late [9,10,11,12,13].

With the development of nanotechnology, porous materials, for which cavities represent more than 30% of the total volume, have recently been developed. Representative porous materials include active carbon and zeolite, and studies on their use as construction materials have been reported [14,15,16]. However, active carbon increases environmental hazards, such as fine dust, owing to logging and heating. In the case of zeolite, only mordenite and clonoptilolite are functional among the entire zeolite mineral groups, but their reserves in South Korea are small [17].

Feldspar, a representative aluminosilicate mineral, is a commonly found mineral as it accounts for 60% of Earth’s crust. It is used for the manufacture of glass in addition to various potteries and ceramics, and is also used for non-functional purposes, such as land reclamation [18,19]. In South Korea, feldspar is mainly extracted from granite and quartz bedrock, and its reserves are abundant; thus, it is available at low cost.

For feldspar, the mineral composition and surface structure are changed by the weathering process. Cavities are observed on the surface of weathered feldspar porphyry, showing a porous structure. Especially, in feldspar phenocrysts, tens of thousands to hundreds of thousands of cavities are observed. The formation of cavities is related to the specific surface area and the cation exchange capacity [20]. Therefore, feldspar with a porous structure is expected to improve the physical and chemical characteristics, such as adhesion to cement, heat transfer, and preservation capabilities.

In the floor structure of a typical Korean house, cushioning or insulation materials (more than 20 mm) and lightweight foamed concrete (more than 40 mm) are placed on the concrete slab and then hot water pipes are installed, as shown in Figure 1. The heat storage layer (more than 40 mm) composed of sand and cement is then constructed on the top [21]. In this study, a certain proportion of cement was replaced with porous feldspar in the heat storage layer to increase the thermal efficiency of floor heating. Thermal, mechanical, and chemical methods were used for the activation of natural feldspar, and changes in the density, strength, and surface structure were observed. For the utilization of porous feldspar as a flooring material, it was mixed with cement and the strength characteristics were evaluated according to the mixing ratio. In addition, the thermal conductivity and heat storage characteristics were monitored through the test construction to evaluate porous feldspar as a substitute for cement.

## 2. Materials and Methods

### 2.1. Materials

Table 1 shows the mineral and chemical compositions of the porous feldspar used in this study. X-ray fluorescence (XRF) analysis was conducted with samples from three areas in South Korea. The analysis results showed that the content of two components, i.e., SiO_2_ and Al_2_O_3_, accounted for more than 80%. Figure 2 shows the scanning electron microscope (SEM, VEGA3 SBH, TESCAN, Brno, Czech Republic) image of weathered feldspar. It can be observed that irregular cavities are present on the surface and they are connected to each other. When the pore distribution of porous feldspar was measured, a high specific surface area of 334.5 m^2^/g was obtained. The measurement was performed using TriStar TM as an analyzer (TriStarTM II 3020, Micromeritics, GA, USA) and the Brunauer–Emmett–Teller (BET) method. It has been reported that materials with porous structures have excellent physical and chemical characteristics owing to the high specific surface area and the cation exchange capacity. Therefore, it was judged that the pore characteristics of the feldspar used in this study satisfied the above characteristics. As for the materials used in the experiment, rocks with developed feldspar phenocrysts were collected from the granite and diorite rocks in the Chung-ju area, and were used in their powder form.

### 2.2. Experimental Conditions

To examine the reactivity of porous feldspar with cement, the uniaxial compressive strength according to the mixing ratio was measured first and the results were compared with the compressive strengths of other substitute materials. Furthermore, when cement is replaced with porous feldspar, a reduction in strength is expected. To compensate for the reduction, mechanical activation for reducing the particle size of materials and chemical activation for improving chemical reactions by mixing a solidifying agent were employed. In addition, mixing tests were conducted to evaluate the applicability of porous feldspar to the heat storage layer by replacing cement and sand. The purpose and method of each test are as follows.

#### 2.2.1. Characteristics of Strength

The purpose of this test was to investigate the strength characteristics according to the weight ratio of porous feldspar powder. The mixing ratio of cement and porous feldspar was varied, and three specimens were fabricated for each mixing ratio in accordance with KS L ISO 679 [22]. The specimens were cured at room temperature (20–24 °C) and humidity (50–60%) for 7 days. The average compressive strength of the three specimens was used as the compressive strength under each condition (EXP-R1 to EXP-R10). Table 2 shows the reactivity test of cement and feldspar.

Industrial minerals that can be widely utilized as construction materials are clay minerals generated from weathered silicate minerals [23]. Among silicate minerals, silica fume, metakaolin, illite, and dolomite, which are representative pozzolanic materials containing a large amount of silica and alumina and are highly reactive with cement, were selected as substitutes for cement. When cement was replaced with porous feldspar, the compressive strength was compared with those of these materials (EXP-RM to EXP-RF). Five specimens were fabricated under each condition in accordance with KS L ISO 679, and the average compressive strength was used. After the fabricated specimens were cured at room temperature for three days, the uniaxial compressive strength was measured. Table 3 shows the mixing ratios of the materials under the above experimental conditions.

#### 2.2.2. Method of Activation and Experimental

To compensate for the strength reduction when porous feldspar was used as a substitute for cement, mechanical and chemical activation methods were used. The particle size of the material causes changes in the physical characteristics, such as the unit weight and compressive strength. The unit weight of a material can change its thermal diffusion and heat storage characteristics. In this study, the unit weight of materials for each particle size was measured in accordance with ASTM C 128 (KS F 2504) for mechanical activation [24,25]. In this instance, all the tests were conducted five times to improve objectivity. The compressive strength test was then conducted on mortar, in which porous feldspar with various particle sizes was substituted for sand. The specimens used in the experiment were fabricated and the compressive strength test was conducted in accordance with KS L ISO 679. Section 1 in Table 4 shows the test conditions for mechanical activation.

When feldspar was used as a material to replace cement, a chemical activation method involving mixing a solidifying agent was used for preserving strength. A developed liquid-type inorganic product was used as a solidifying agent in the test. Section 2 in Table 4 shows the test conditions for chemical activation. The fabrication of specimens and the compressive strength test were conducted in accordance with KS L ISO 679, and changes in the surface structure were observed using SEM imaging (EXP-A1 to EXP-A3). The particle size of the feldspar used for specimen fabrication was 80 µm, which was selected based on the strength change results obtained via mechanical activation.

#### 2.2.3. Strength Test of Feldspar and Mortar

This test was conducted to investigate the strength characteristics when cement and sand were replaced with porous feldspar. Specimens in which the ratio of ordinary cement to sand was 1:3 (EXP-FM1) and other specimens in which sand was replaced with feldspar smaller than 1 mm and feldspar powder smaller than 40 µm (EXP-FM2) were fabricated in a cubic form (side length: 50 mm). Three specimens were fabricated under each condition. After they were cured in water for 3–28 days, the compressive strength was measured in accordance with the ASTM C109/C109M (KS L 5105) method [26,27]. Table 5 shows the material mix and experimental conditions.

#### 2.2.4. Thermal Diffusion and Heat Storage Test

In the thermal diffusion and heat storage test, a temperature sensor was embedded at the center of 50 × 50 × 50 mm specimens, which were then subjected to water curing for 28 days [28]. The mixing proportions of the specimens were the same as those in EXP-FM1 and EXP-FM2, as shown in Table 5. The fabricated specimens were installed on top of a 400 × 400 mm hot plate, as shown in Figure 3. Their temperatures were measured every minute using a data logger. The test was conducted under two conditions. First, the temperature of the hot plate was set to 100 ℃ after installing the two specimens on top of the hot plate. Subsequently, the specimens were separated from the hot plate after a 60 min heating period. They were then cooled at room temperature (22–24 °C) to investigate their thermal diffusion effects and heat storage characteristics. In the second method, the thermal diffusion characteristics were investigated by repeating the heating and cooling periods to simulate conditions similar to those of the actual floor heating. Heating and cooling periods were repeated for 300 min at 30 min intervals.

Floor heating circulates water heated to a high temperature through a pipe embedded in the heat storage concrete. In this instance, hot water is repeatedly supplied according to the set temperature. The second method is similar to this process.

### 2.3. Pilot Test

To evaluate the thermal conductivity, heat storage characteristics, and energy efficiency of porous feldspar, a large-scale single-story experimental building composed of temporary structures was installed. The outer wall of the experimental building was insulated to reduce the influence of the external temperature. Inside this experimental building, two temporary houses each of dimensions 3000 (L) × 4000 (W) × 3000 (H) mm were constructed. Figure 4 shows the design drawings of the heat storage experiment. Based on the floor layer construction standard in Figure 1, a heat storage layer composed of typical concrete mortar (PT-1) was constructed in a temporary house and a heat storage layer in which sand was replaced with feldspar (PT-2) was installed in the other temporary house. EXP-FM1 and EXP-FM2 were applied as the concrete mixing ratios of the heat storage layers. In addition, the temporary houses were separated from the ground by 50 cm to minimize the influence of the ground temperature. Construction and measurement for the two conditions were simultaneously performed to minimize the influence of external environmental factors. After installing a 2 kW electric boiler in each temporary house for hot water supply, a watt-hour meter was installed to determine the power consumption according to the experimental conditions. The temperature of the heat storage layer was measured using an infrared thermal-imaging camera (FLIR A615, temperature range: 20–150 °C, measurement error: 1 °C) (FLIR Systems Srl, Milan, Italy). Temperature sensors were installed at intervals of 80 cm in the heat storage layer to measure the temperature change and power consumption owing to the boiler operation. The measured data were transmitted to the Internet via a wireless router and stored in a cloud service to apply a remote measurement method [29]. The power consumption during the boiler operation was calculated from the images obtained by the CCTVs installed in the watt-hour meters [30].

## 3. Results and Discussion

### 3.1. Response Characteristics

In the development of substitute materials for cement, the cement replacement rate is generally determined by the uniaxial compressive strength. The strength decreases if the mixing proportions of substitute materials are excessive, and the cement content increases if they are too low. In other words, appropriate mixing of cement and substitute materials is important because increasing the cement content is beneficial for strength but not for the environment.

Figure 5 shows the uniaxial compressive strength according to the mixing ratio of cement and porous feldspar powder. The compressive strength of the specimen with only cement was 11.43 MPa, and the compressive strength decreased as the cement content decreased. The compressive strength linearly decreased for EXP-R10 to EXP-R4 in which the cement content was reduced to 40%, and it rapidly changed for EXP-R3 to EXP-R1 in which the cement content was less than 30%. This indicates that the proper mixing proportion of porous feldspar powder is less than 70% for a modest reduction in strength. 

Figure 6 shows the uniaxial compressive strength according to the silicate mineral type. When feldspar powder was used, the strength was approximately 16–63% higher compared with that when other silicate minerals were used, indicating that porous feldspar can be used as a substitute for cement. As for the characteristics of aluminosilicate minerals, SiO_2_ and Al_2_O_3_ are representative pozzolanic components. It appears that porous feldspar increased the strength through the reaction with Ca(OH_2_) in the cement hydration process because approximately 80% of its content is accounted for by these two components.

### 3.2. Mechanical Activation

Figure 7 shows the unit weight and compressive strength according to the particle size of porous feldspar. The unit weight decreased as the particle size of the feldspar decreased. The unit weight for the particle size of 20 µm was 1.06 g/cm^3^, which was approximately 31% lower than that for 500 µm. As for the compressive strength according to the particle size, the lowest strength was observed for the largest particle size of 200 µm. As the particle size decreased, the strength slowly increased up to approximately 90%, confirming that the physical characteristics were improved by reducing the particle size through mechanical activation.

### 3.3. Chemical Activation

Figure 8 shows the compressive strength according to the mixing condition of porous feldspar. The compressive strength of EXP-A2 in which 100% cement was mixed with solidifying agent corresponding to 0.1% of the cement weight ranged from 15 to 19 MPa, showing that the compressive strength was improved by approximately 33% compared with that of EXP-A1 in which only 100% cement was used. The compressive strength of EXP-A3 in which 70% of the cement weight was replaced with porous feldspar and solidifying agent corresponding to 0.1% of the cement weight added ranged from 15 to 18 MPa, which was approximately 30% higher than that of EXP-A1 in which only cement was used. Figure 9 shows the surface structures of the specimens analyzed using SEM. For EXP-A3 in which cement was replaced with porous feldspar, reaction products of the chemical reactions of the inorganic solidifying agent and porous feldspar were observed. Table 6 shows the results of analyzing (SEM-EDS, TESCAN VEGA3 SBH) the chemical compositions of the specimens used in the tests. Na and Cl, which are the major components of the solidifying agent, were detected in EXP-A2 and EXP-A3, in which the solidifying agent and porous feldspar were added. In EXP-A3, the Si and Al contents were 29.6% and 7.2%, respectively, values two to three times higher than those of the other samples. They appear to have increased the strength through the reaction with Ca(OH)_2_ generated from the cement hydration process.

### 3.4. Evaluation of Substitute Materials

The compressive strength test was conducted to evaluate the applicability of porous feldspar to the heat storage layer in the heating floor layer. For a relative comparison, a specimen was fabricated in the same manner by mixing cement mortar and sand in a ratio of 1:3. The mixing proportion of porous feldspar less than 70% was proposed in Section 3.1, and the inorganic solidifying agent corresponding to 0.1% of the cement weight was added based on the experiment results in Section 3.3 to prevent the rapid reduction in strength and to increase the addition of porous feldspar.

Figure 10 shows the results of the compressive strength test. As for the strength characteristics according to the curing time, the strength showed a tendency to increase over time for both materials. Especially, for EXP-FM2 with porous feldspar, the compressive strength at 3 days was 10.53 MPa even though the cement content was reduced by 5%. This result indicates that the strength was improved by approximately 43% compared with that of EXP-FM1. The compressive strengths of EXP-FM1 and EXP-FM2 at seven days were 7.3 and 14.94 MPa, respectively, showing that the strength of EXP-FM2 was two times higher. Both compressive strengths satisfied the quality criterion of South Korea (strength at seven days: 7 MPa) [31]. The compressive strengths of EXP-FM1 and EXP-FM2 at 28 days were 14.14 and 18.97 MPa, respectively, confirming that the strength of EXP-FM2 with porous feldspar was approximately 35% higher than that of EXP-FM1 even though its strength increment slightly decreased.

### 3.5. Characteristics of Thermal Diffusion and Heat Storage

Figure 11 shows the temperature changes in the EXP-FM1 and EXP-FM2 specimens when they were heated on the heating plate for 60 min and then cooled at room temperature (22–24 °C) for 150 min. The maximum temperature of the EXP-FM2 specimen, which replaced the sand with porous feldspar, was 66.4 °C. This was 7 °C higher than the maximum temperature (59.4 °C) of EXP-FM1, a comparison target, confirming the high thermal diffusivity of the specimen containing the porous feldspar. In contrast, when cooled at room temperature after heating for 60 min, EXP-FM2 exhibited a sharp decrease in temperature at the beginning of the cooling period and maintained an approximately 1.3 °C higher temperature than EXP-FM1. This could be because the heat loss in EXP-FM2, which exhibited relatively higher temperatures, was more, owing to the equilibrium between the temperature inside the specimen and the outside temperature during the cooling period at room temperature.

Figure 12 shows the results of the test in which the heating and cooling periods were repeated at 30 min intervals. The maximum temperatures during the heating period were approximately 59 °C for EXP-FM1 and 65 °C for EXP-FM2, resulting in a difference of approximately 6 °C. Contrarily, the minimum temperatures during the cooling period were approximately 45 °C for EXP- FM1 and 48 °C for EXP-FM2, resulting in a difference of approximately 3 °C. As the test was repeated, the maximum and minimum temperatures exhibited similar values.

In the cooling process, EXP-FM2 had higher temperatures than EXP-FM1 unlike the results shown in Figure 11. This is because a method similar to the actual heating process was selected in the cooling process without separating the specimens from the heating plate. A point to be noted from these results is whether the created mortar, in which the sand was replaced with porous feldspar and the cement content was decreased, is suitable as a heat storage layer in flooring material. In Section 3.4, it was confirmed that the specimen containing the porous feldspar satisfied the strength criterion for the heat storage layer. In addition, the results of the thermal diffusion and the heat storage test confirmed that EXP-FM2, which replaced the sand with porous feldspar, had faster thermal diffusion and better heat storage characteristics than EXP-FM1, a comparison target. In general, low density of concrete causes high thermal diffusivity but leads to a low specific heat capacity. In this test, however, EXP-FM2, which had a lower density, was superior to EXP-FM1 in terms of both heat transfer and heat storage. It appears that the heat storage effect was high owing to the porosity characteristics of feldspar, a substitute material for sand. 

### 3.6. Pilot Test

#### 3.6.1. Characteristics of Thermal Conductivity and Heat Storage

Figure 13 shows the temperature distribution of the heat storage layer measured using the thermal-imaging camera during the heating time after hot water was supplied to the floor. The temperature of the supplied hot water was 55 °C. At 10 min after the boiler operation, the heat storage layer mixed with porous feldspar (PT-2) exhibited a temperature approximately 2 °C higher than that of the heat storage layer composed of ordinary concrete (PT-1). The temperature difference increased to approximately 3 °C at 2 h after the boiler operation, indicating that PT-2 had better heat transfer than that of PT-1. Figure 14 shows the temperature variation during the cooling period after the boiler operation was stopped. PT-1 exhibited a fast cooling speed and no thermal image could be obtained after 90 min. The average temperature was 21 °C after 120 min and 18 °C after 180 min, which was identical to the temperature before the boiler operation. In contrast, PT-2 with porous feldspar exhibited temperatures approximately 2 °C higher than those of PT-1. The temperature remained at over 23 °C even after 180 min of cooling, confirming the excellent heat storage characteristics.

These results can be divided into the influence of the material properties of the heat storage layer and that of the bottom storage layer structure (Figure 1). First, in terms of material properties, the concrete containing the porous feldspar (PT-2) was favorable for thermal diffusion because its density was lower than that of PT-1, as shown in the thermal diffusion and heat storage test in Section 3.5. In contrast, PT-1 exhibited slow thermal diffusion because the density of its heat storage layer was higher compared to that of PT-2. Owing to the high thermal resistance, more heat flow occurred in the bottom direction (lightweight foamed concrete + slab), thereby causing relatively low thermal diffusion to the heat storage layer.

#### 3.6.2. Energy Efficiency

Figure 15 shows the heat storage layer temperature and power consumption during the experiment period under the two conditions. PT-2 exhibited a higher temperature increase rate than that of PT-1 until 100 min after heating, and the increase rate decreased after 30 °C. PT-1 showed a temperature increase rate similar to that of PT-2 until 20 °C, but its temperature increase rate became lower than that of PT-2. During 180 min of heating, the maximum temperature was 34.4 °C for PT-2 and 31.6 °C for PT-1, showing that the value for PT-2, which used porous feldspar, was 2.7 °C higher. When the target temperature of the heat storage layer was set to 25 °C, the boiler operation was stopped after approximately 40 min and the power consumption was 1.84 kWh for PT-2. For PT-1, the boiler operation was stopped after approximately 70 min and the power consumption was 3.22 kWh. In other words, the boiler was operated 30 min longer for PT-1 than for PT-2, and PT-1 required the additional power of 1.38 kWh, confirming that PT-2 could save approximately 57% power. If the boiler restart temperature is set to 20 °C after the cooling process, it is expected that the boiler will restart after 380 min for PT-2 and after 310 min for PT-1; thus, PT-2 will delay the boiler operation by approximately 70 min. Through the experiment, the rapid temperature increase effect in the heating process owing to the high thermal conductivity caused by the specific surface area characteristics of porous feldspar and the energy-saving effect in the cooling process owing to the delay effect caused by the heat storage characteristics of porous feldspar could be confirmed.

## 4. Conclusions

In this study, feldspar, which is found in South Korea in large quantities and has a porous structure owing to weathering, was used as a substitute for sand in cement. When sand was replaced with porous feldspar and four other silicate minerals in the cement mortar, the specimen that used the porous feldspar exhibited approximately 16–63% higher compressive strength, thereby confirming a higher reactivity with cement than other minerals.

When the particle size was reduced via mechanical activation to increase the specific surface area of porous feldspar, the unit weight decreased by approximately 30%, but the uniaxial compressive strength increased by up to 90%, confirming that the physical characteristics were improved.

A solidifying agent was mixed in to compensate for the strength reduction caused by the addition of porous feldspar. When 70% of the sand weight was replaced with porous feldspar and the solidifying agent was mixed in, the compressive strength was improved by approximately 30% compared with when only cement was used.

When chemical activation (2:8) was performed after reducing the cement content by 5% and replacing the sand with porous feldspar, the compressive strength at 3 days improved by approximately 43% compared to that of the specimen in which cement and sand were mixed in a ratio of 1:3. The compressive strength at 7 days was approximately two times higher.

As for the thermal diffusion, the mortar in which the sand was replaced with the porous feldspar exhibited approximately 6–7 °C higher temperatures than that of the standard concrete in the heating process. In addition, it maintained approximately 1–3 °C higher temperatures in the cooling process. This was because the mechanical activation increased the thermal diffusion by reducing the density of the porous feldspar mortar and the heat storage effect of the feldspar was relatively better owing to its porosity.

In a large-scale model experiment, porous feldspar exhibited excellent thermal diffusion and heat storage characteristics in the heat storage layer as well as an approximately 57% energy saving effect, thereby confirming its high applicability as a heat storage layer material.

The use of porous feldspar as a substitute material for aggregate can reduce the cement content, thereby decreasing the CO_2_ emissions. In addition, porous feldspar is an economical option owing to its easy availability and inexpensive characteristics.

## Figures and Tables

**Figure 1 materials-13-04204-f001:**
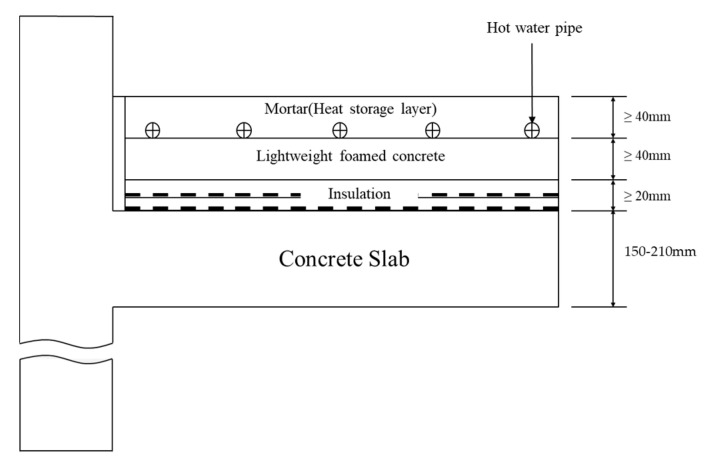
Construction standard of the bottom layer in Korea.

**Figure 2 materials-13-04204-f002:**
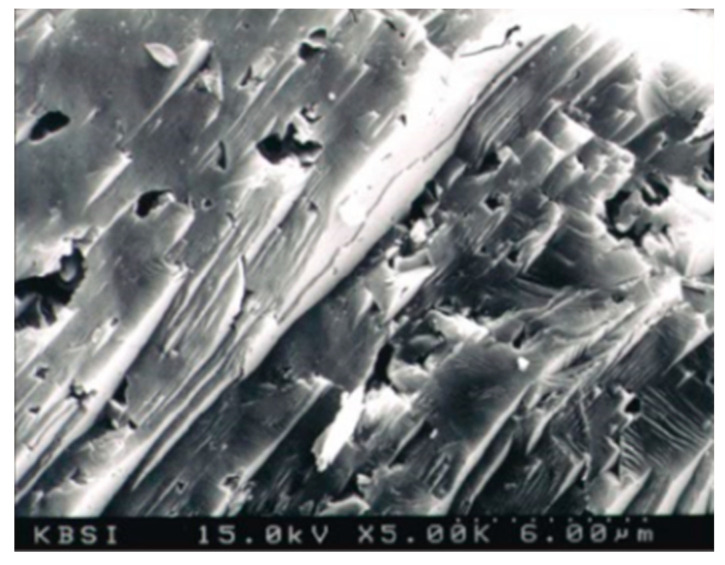
SEM image of weathered feldspar.

**Figure 3 materials-13-04204-f003:**
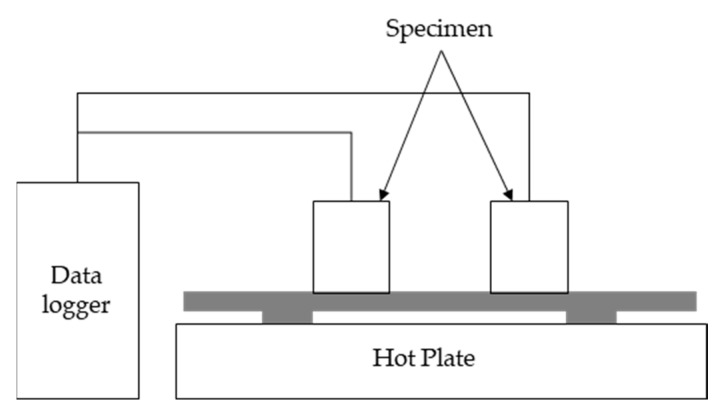
Thermal diffusion and heat storage test device.

**Figure 4 materials-13-04204-f004:**
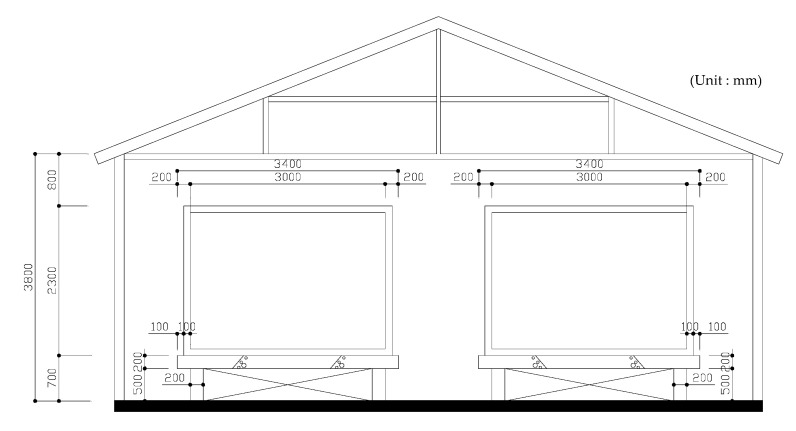
Design drawings of heat storage experiment.

**Figure 5 materials-13-04204-f005:**
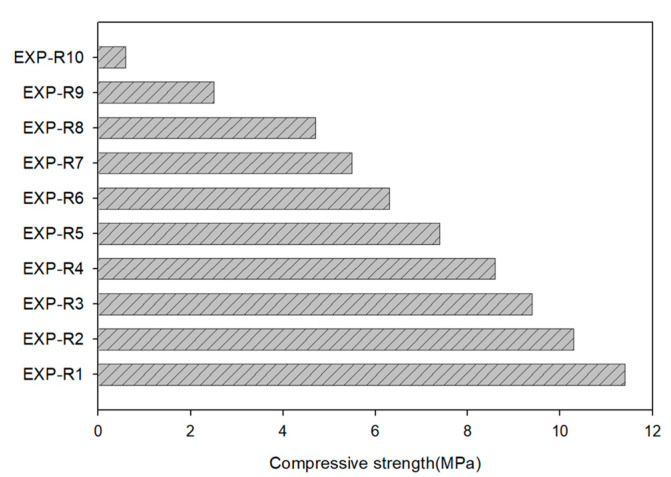
Compressive strength according to the ratio of cement and porous feldspar.

**Figure 6 materials-13-04204-f006:**
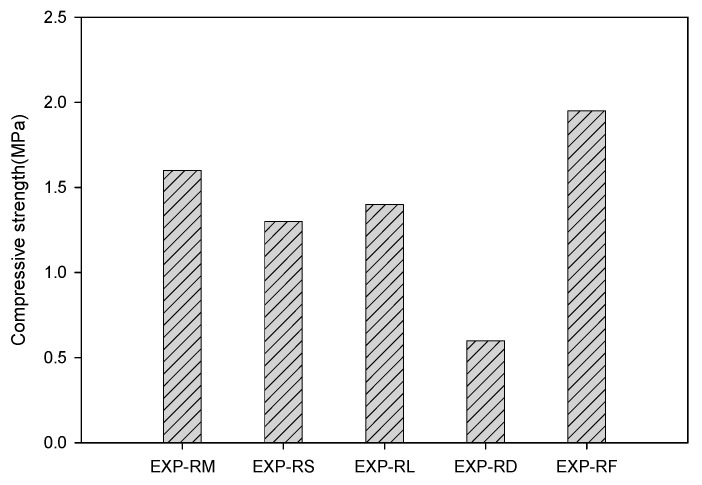
Compressive strength according to the type of silicate minerals.

**Figure 7 materials-13-04204-f007:**
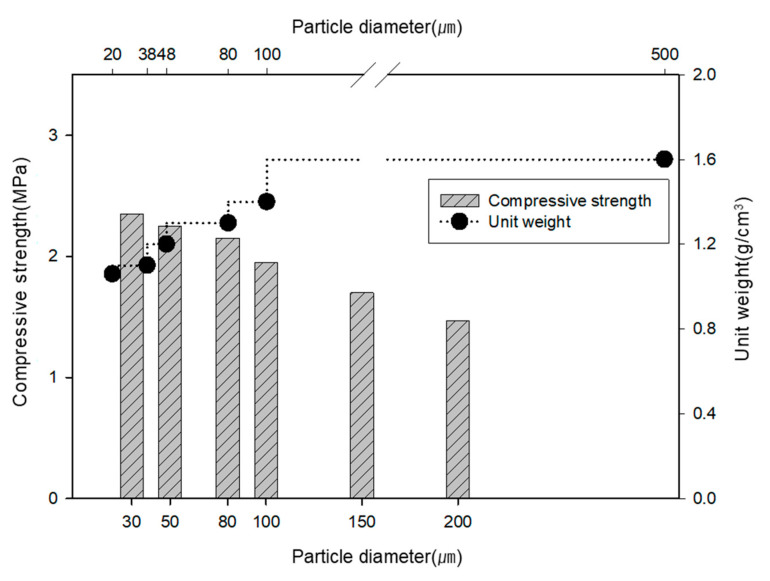
Relationship between the unit weight and compressive strength of feldspar according to the particle size.

**Figure 8 materials-13-04204-f008:**
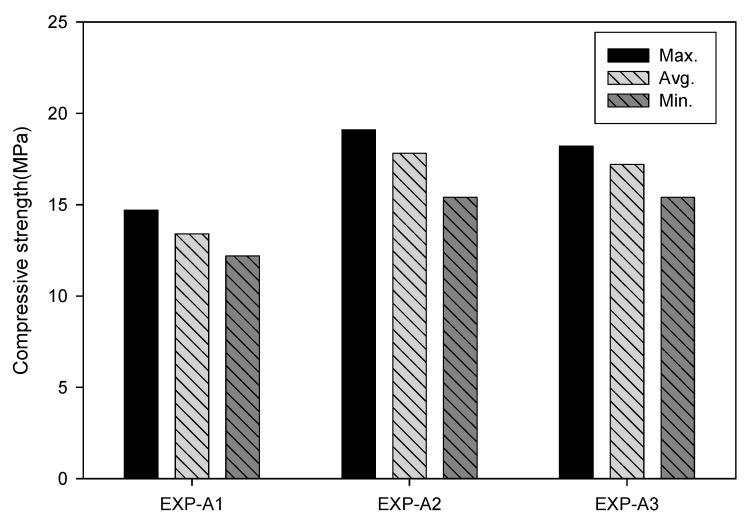
Compressive strength of cement and feldspar mixture.

**Figure 9 materials-13-04204-f009:**
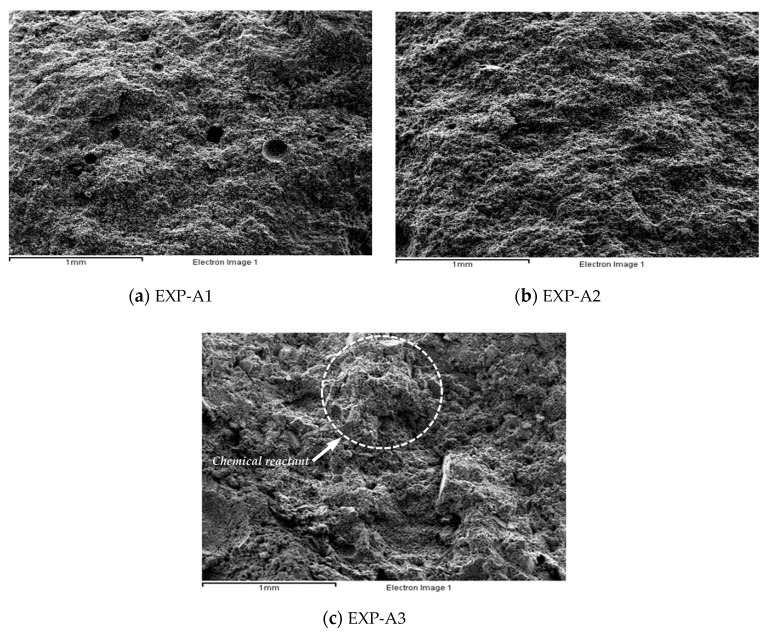
SEM image of specimens. (**a**) EXP-A1, (**b**) EXP-A2, and (**c**) EXP-A3.

**Figure 10 materials-13-04204-f010:**
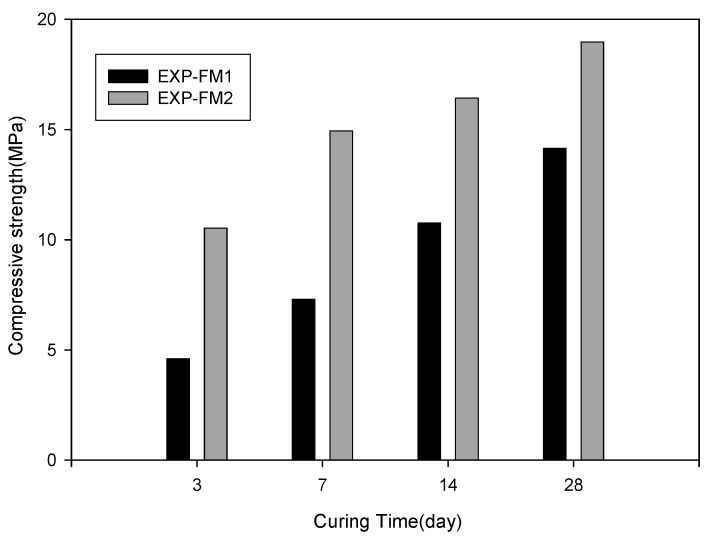
Compressive strength for different mixing ratios and curing times.

**Figure 11 materials-13-04204-f011:**
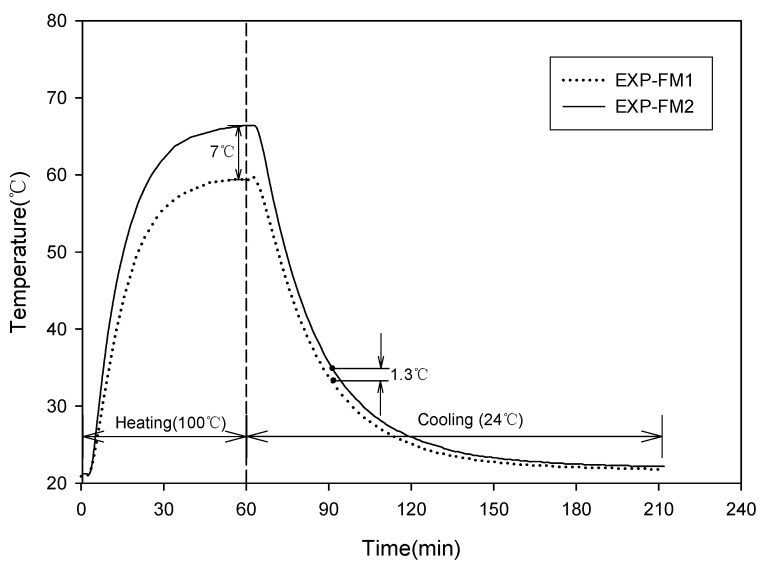
Result of heat storage (heating for 60 min).

**Figure 12 materials-13-04204-f012:**
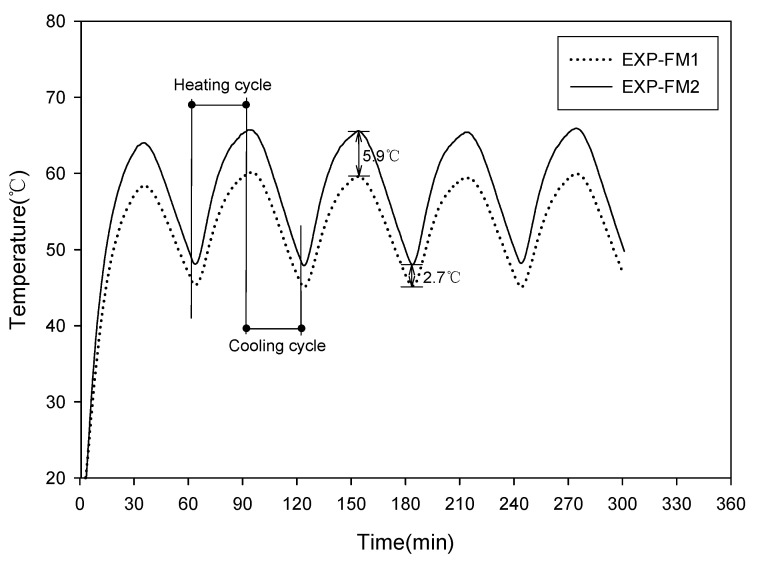
Characteristics of heat diffusion and storage (repeated heating).

**Figure 13 materials-13-04204-f013:**
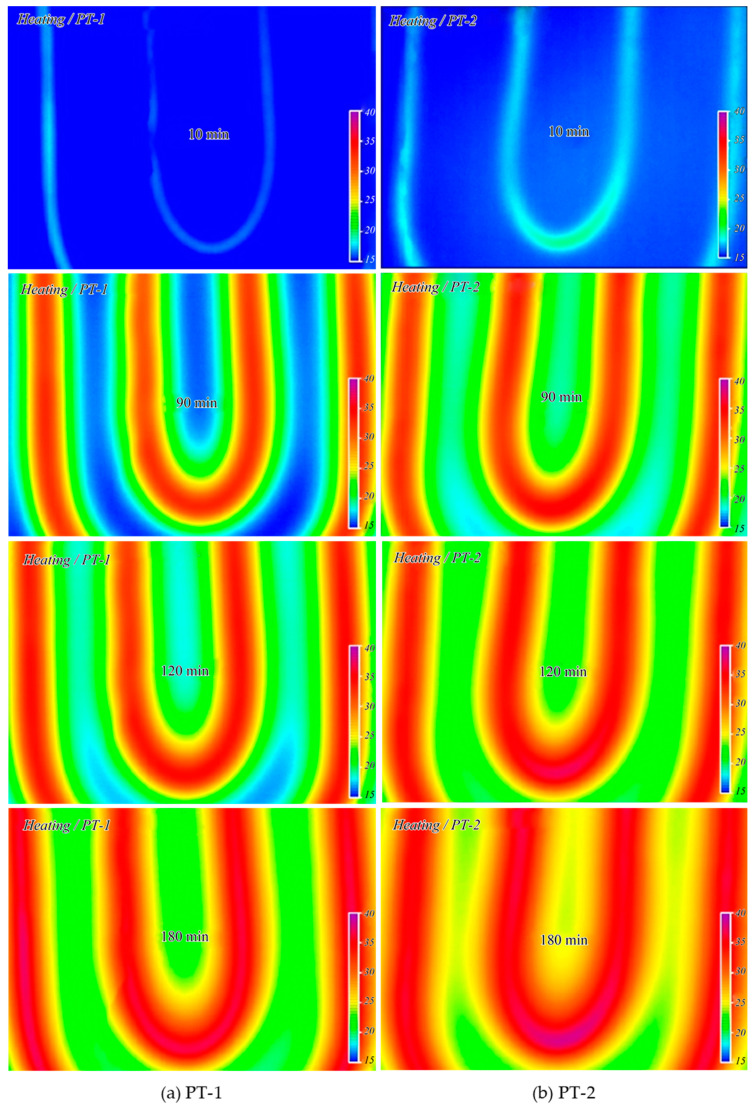
Temperature change of heat storage layer (heating cycles).

**Figure 14 materials-13-04204-f014:**
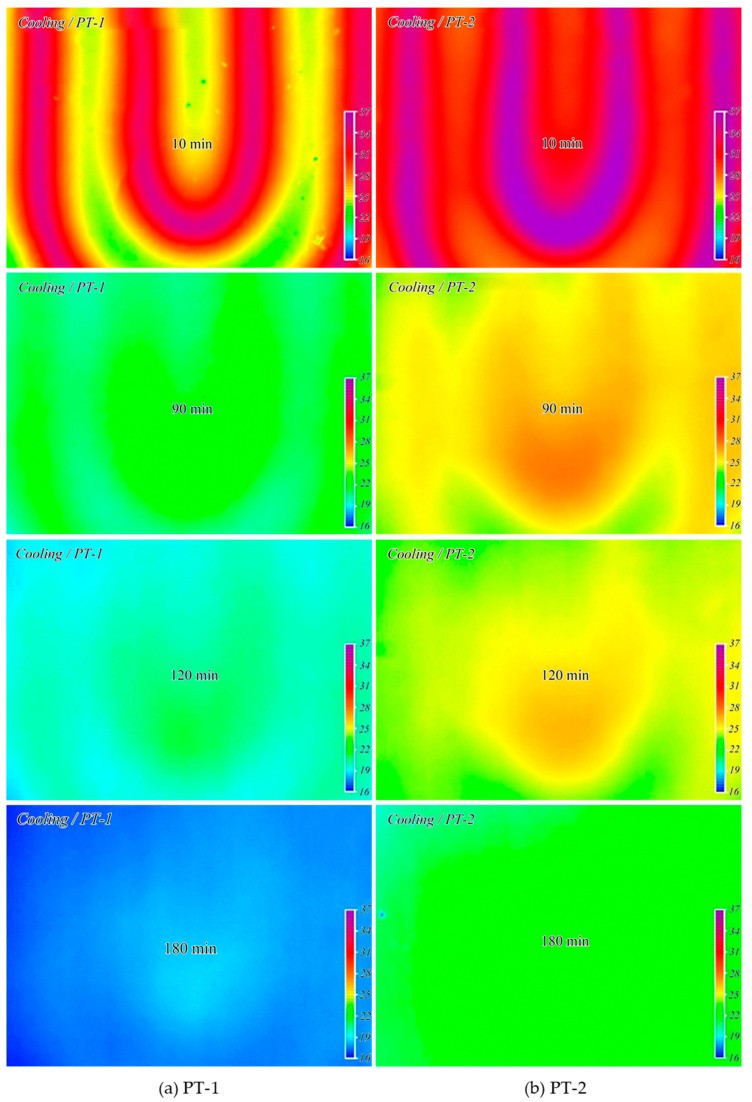
Temperature change of heat storage layer (cooling cycles).

**Figure 15 materials-13-04204-f015:**
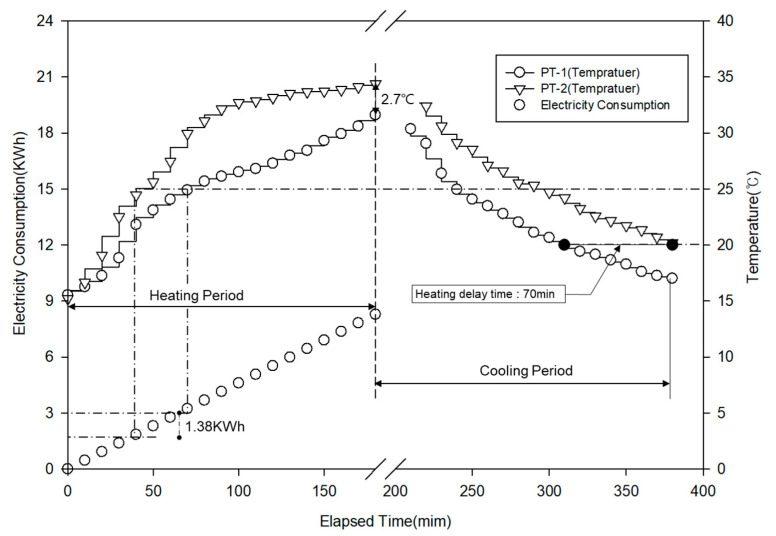
Change of temperature and electric consumption.

**Table 1 materials-13-04204-t001:** Characteristics of experimental materials (porous feldspar).

I. Mineral Composition of Feldspar Porphyry										
Albite (NaAlSi_3_O_8_)			Quartz (SiO_2_)			Orthoclase (KAlSi_3_O_8_)			Chlorite	
(%)										
38.8			26.5			22.7			8.3	
**II. Chemical Composition of Feldspar**										
Location	SiO_2_	Al_2_O_3_	K_2_O	Na_2_O	CaO	Fe_2_O	MgO	TiO_2_	LOI	Other
	(%)									
Chung-ju	69.59	13.07	2.7	4.53	2.56	2.49	1.61	0.49	2.46	0.50
Muan	67.10	15.26	5.02	3.84	1.88	3.34	0.73	0.40	1.59	0.84
Namwon	68.95	14.59	4.63	3.39	1.29	3.03	0.67	0.37	2.71	0.37

**Table 2 materials-13-04204-t002:** Test conditions for reactivity of cement and feldspar.

Mixed Ratio (Cement: Feldspar)										Feldspar Size	Curing Time	W/C Ratio
EXP-R1	EXP-R2	EXP-R3	EXP-R4	EXP-R5	EXP-R6	EXP-R7	EXP-R8	EXP-R9	EXP-R10	20–500 µm	7 day	0.5
10:0	9:1	8:2	7:3	6:4	5:5	4:6	3:7	2:8	1:9			

**Table 3 materials-13-04204-t003:** Test conditions for reactivity of cement and silicate minerals.

Mixed Ratio (Cement: Clay Mineral)	Metakaolin	Silica Fume	Ilite	Dolomite	Feldspar	Curing Time (Day)	W/C Ratio
7:3	EXP-RM	EXP-RS	EXP-RL	EXP-RD	EXP-RF	3	0.5

**Table 4 materials-13-04204-t004:** Test conditions for activation methods (mechanical and chemical activation).

I. Mechanical Activation									
Test Methods	Feldspar Size (µm)						Mixed Ratio (PC:FS) *	Curing Time (day)	W/C Ratio
Unit weight test	20	38	48	80	100	500	0:10	-	-
Compressive strength test	30	50	80	100	150	200	70:30	3	0.5
**II. Chemical Activation**								**Curing time (day)**	**W/C Ratio**
Test No.		EXP-A1		EXP-A2	EXP-A3		Feldspar size (µm)	28	0.5
Mixed ratio (PC:FS:S) *		100:0:0		100:0:0.1	30:70:0.1		80		

* PC: Portland cement (%), FS: Feldspar (%), S: Solidifying agent of 0. 1% by weight of cement (%).

**Table 5 materials-13-04204-t005:** Test condition of substitute materials with feldspar.

**Test Method**	**Mixed Ratio**		**Curing Time (Day)**	**W/C Ratio**
	EXP-FM1 (PC:AG) *	EXP-FM2 (PC:AGF:PS:S) *	3, 7, 14, 28	0.5
Compressive strength test	25:75	20:40:4%:0.1		

* PC: Portland cement (%), AG: sand (%), AGF: Feldspar ≤ 1000 µm (%), PF: Feldspar ≤ 40 µm (%), S: Solidifying agent of 0.1% by weight of cement (%).

**Table 6 materials-13-04204-t006:** SEM-EDS elemental composition of specimens (EXP-A1, EXP-A2, EXP-A3).

	Component	Si	Al	Ca	Na	Cl	Mg	K	S	Fe
Sample No.		(%)								
EXP-A1		14.1	2.9	73.8	N.D *	N.D *	1.9	1.2	3.5	2.6
EXP-A2		11.2	3.1	73.6	0.8	0.9	1.7	3.2	2.7	2.8
EXP-A3		29.6	7.2	43.9	1.6	1.3	3.1	4.7	2.5	6.1

* N.D: Non-detection.

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
