# Peer review of "Characteristics of CO2 and Energy-Saving Concrete with Porous Feldspar"

_materials, 2020, doi:10.3390/ma13184204_

Round 1
Reviewer 1 Report
Figure 2. SEM image of weathered feldspar
Table4: the units for percentage should same, maybe wt% for all (same for all)
Unit for Curing time (maybe days)
L 171 of 50 X 50 X 50 mm3 (superscript) same for the rest
L243 !!! wh1`ich
Fig 7 MPa
Figure 9. “For EXP-A3 in which cement was replaced with porous feldspar, reaction products of the chemical reactions of the inorganic solidifying agent and porous feldspar were observed” can you point on the figure where these reaction products were observed?
Reviewer 2 Report
Strong Points:
Authors described the feldspar usage as a substitute for sand, sand was replaced with porous feldspar and four silicate minerals in the cement mortar, and analyses were performed on chemical activation, mechanical parameters (like compressive strengths), heat storage characteristics. They claimed that porous feldspar is an economical option owing to its easy availability and inexpensive characteristics with decreasing the CO2 emissions.
Weak Points:
- Please explain the differences of these 3 areas where you got the 3 samples. In addition, there is no information on "Muan" and "Namwon" in the text. Please explain the differences between them, and/or why you chose those place? If you chose those areas specifically, could you please share the characteristics?
- In page 1 line 34; please explain LNG and LPG.
- Please give a link or reference to "KS L ISO 679".
- Typing errors. e.g., "...50 mm3 specimens..." in Page6 Line171.
- Figures 1 and 3 should be improved. They look huge. Tables can be fitted well.
- The newest reference belongs to "2016" in this manuscript. Please add the latest reference reports.
Besides these; Manuscript should provide sufficient background. Please include all relevant references.
There are some examples where reference reports are crucial:
- In Page1, Line 36; Please share reference articles to the comments on the health risks of cement on humans.
- In Line 39; share refs for "...efforts are being made to reduce greenhouse gas emissions."
- In Page1, Line 40; Please share references for the comment on "...emits 700 kg of carbon dioxide per ton..."
- In Page2, Line 51; please share reference for the comment on the "...active carbon increases environmental hazards, such as fine dust, owing to logging and heating."
- Reference 17 should be after "The formation of cavities is related to the specific surface area and the cation exchange capacity." in Page2, Line62.
